# First report of a major management target species, chironomid *Paratanytarsus grimmii* (Diptera: Chironomidae) larvae, in drinking water treatment plants (DWTPs) in South Korea

Jae-Won Park[1☯], Kiyun Park[2☯], Ihn-Sil Kwak[1,2]*

**1** Department of Ocean Integrated Science, Chonnam National University, Yeosu, Korea, **2** Fisheries Science Institute, Chonnam National University, Yeosu, Korea

☯ These authors contributed equally to this work.
* inkwak@hotmail.com, iskwak@chonnam.ac.kr

**Data Availability Statement:** All relevant data are within the paper and its Supporting Information files.

## Abstract

Ensuring the supply of safe and high-quality drinking water can be compromised by the presence of chironomid larvae in drinking water treatment plants (DWTPs), which may contaminate municipal water systems through freshwater resources. Chironomids are dominant species known for their resilience to a broad range of extreme aquatic environments. This study aimed to identify the morphological characteristics and obtain genetic information of the chironomid *Paratanytarsus grimmii* found in the water intake source and freshwater resource of DWTPs in Korea, highlighting the potential possibility of a parthenogenetic chironomid outbreak within DWTP networks. The distribution of chironomid larvae at the water intake source site (DY) of the Danyang DWTP and the freshwater resource (ND) of the Nakdong River was investigated. A total of 180 chironomid individuals, encompassing three subfamilies and six species from six 6 genera were identified at the DY site, with *Procladius nigriventris* being the dominant species. At the ND site, fifty chironomid individuals, encompassing two subfamilies and six species from six genera, were identified, with *Cricotopus sylvestris* being the dominant species. The morphological characteristics of the head capsule, mentum, mandible, and antennae of six *P. grimmii* larvae collected from the DY and ND sites were characterized. DNA barcoding and phylogenetic analysis revealed distinct mitochondrial diversities between the *P. grimmii* larvae from DY and those from ND. These results provide crucial information for the morphological identification and DNA barcoding of the key management target chironomid *P. grimmii* larvae, which can be used to detect the occurrence of this chironomid species in DWTPs.

## Introduction

Drinking water treatment plants (DWTPs) are crucial for supplying safe and high-quality drinking water, which is essential for public health [1]. These plants treat surface waters from

**Funding:** This work was supported by a grant from the National Institute of Biological Resources (NIBR), funded by the Ministry of Environment (MOE) of the Republic of Korea (NIBR202220202) and the National Research Foundation of Korea, South Korea, funded by the Korean Government (NRF-2018-R1A6A1A-03024314), as well as the Korean Environment Industry & Technology Institute (KEITI) through the Aquatic Ecosystem Conservation Research Program funded by the Korean Ministry of Environment (MOE) (2021003050001). The funders had no role in study design, data collection and analysis, decision to publish, or preparation of the manuscript.

**Competing interests:** No authors have competing interests.

rivers, lakes, groundwater, reservoirs, and streams to provide safe water to municipal areas [2]. Water safety standards mandate the absence of pathogens, organic and inorganic substances, and chemicals in raw water [3, 4]. The treatment processes in DWTPs generally include water intake, aeration, precipitation, biological pretreatment, Pulsator clarification, ozone treatment, sand filtration, chlorination, activated charcoal filtration, microfiltration, clean water storage, and distribution [1, 2]. Water quality is also influenced by the environmental conditions of freshwater bodies, which serve as natural water sources for DWTPs. These sources are open spaces that provide habitats for the breeding of aquatic insects, which can transfer pathogens [5, 6]. Additionally, these sources are increasingly affected by toxic cyanobacterial blooms [7]. Recently, microplastic contaminants have been detected in all types of water sources commonly used by DWTPs [8]. Consequently, the importance of monitoring DWTPs and their freshwater source environments to ensure a clean water supply is growing each year.

Non-biting midge chironomid (Diptera: Chironomidae) larvae are dominant macroinvertebrates and play a crucial role in the food web as a major food source for fish in aquatic ecosystems [9, 10]. These larvae can survive in diverse freshwater environments, from polluted waters to extreme temperature conditions, including in the water and filters of DWTPs [1, 11, 12]. In July 2020, the occurrence of chironomid larvae was first reported in domestic tap water supplied by DWTPs in Incheon, South Korea [13]. The larvae, including *Chironomus flaviplumus*, *Chironomus dorsalis*, *Chironomus kiiensis*, and *Polypedilum yongsanensis*, were found in DWTPs and freshwater sources [1, 13]. In DWTPs in Jeju, chironomid larvae of two species, *Orthocladius tamarutilus* and *Paratrichocladius tammaater*, were detected in faucets, water intake towers, and fire hydrants in DWTPs [14]. The walls of sedimentation spaces in DWTPs provide a habitat conducive to the laying of eggs, which can lead to outbreaks of chironomid larvae in plant effluent [15]. The outbreaks of chironomid larvae in tap water have drawn public attention to the water supply system and the safety of drinking tap water in South Korea. The presence of chironomid larvae in DWTPs has raised concerns not only regarding aesthetic issues but also regarding the safety of drinking water and the potential for pathogen transmission by aquatic macroinvertebrates [4, 10].

*Paratanytarsus grimmii*, a non-biting midge (Diptera: Chironomidae), is a triploid chironomid that reproduces parthenogenetically without males [16, 17]. *Paratanytarsus grimmii* larvae were first discovered in water distribution systems in 1941 [18]. Parthenogenetic species often have the potential to rapidly spread across a wide range of aquatic environments [17]. In DWTPs, the control of *P. grimmii* outbreaks focuses on managing drinking water pipes. The presence of *P. grimmii* in DWTPs poses an indirect threat to public health owing to the production of high levels of organic matter by macroinvertebrate populations, which can promote harmful microbial growth [16, 17]. Although *P. grimmii* has been widely reported in Europe and America [19–21], there have been no reports of this species in DWTPs and raw freshwater sources in Korea to date.

To address this gap in the literature, this study investigated the chironomid community in the water sources of DWTPs in Korea to obtain genetic information and determine the morphological characteristics of chironomid larvae. Additionally, this study also investigated the freshwater resources of Korean DWTPs to determine the presence of *P. grimmii* larvae, a known parthenogenetic species in European water supplies.

## Materials and methods

### Ethical note statement

This research was undertaken in line with the ethical requirements of the Animal Care and Use Committee of Chonnam National University (Yeosu, Republic of Korea). This study did not involve endangered or protected species.

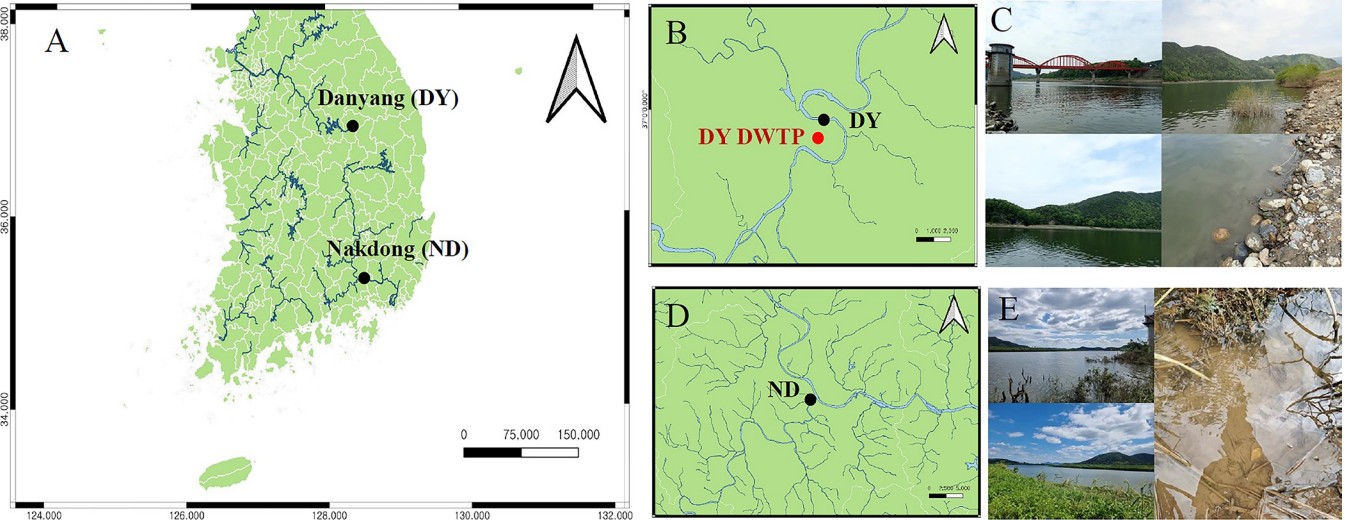

**Fig 1. Chironomid sampling sites in South Korea.** (A) Locations of the water intake source (DY) of the Danyang drinking water treatment plant (DWTP) and the freshwater resource (ND) of the Nakdong River in South Korea. (B) The red circle on the map indicates the location of the Danyang DWTP and the black circle indicates the water intake source of the DWTP. (C) Field photographs of the DY site. (D) The black circle on the map indicates the location of the Nakdong River freshwater resource used for chironomid sampling. (E) Field photographs of the ND site.

## Chironomid sampling and water quality survey in freshwater sources of DWTPs

Chironomid larvae were sampled at two sites: the DY site, the water intake source of Danyang DWTPs (36˚59'35"N 128˚21'27"E), in April 2024, and the ND site (35˚23'26"N 128˚25'45"E), an indirect freshwater resource of the Busan Beomeo DWTP and Samgye DWTP, in April 2023 [1] (Fig 1). The survey of the waterway was conducted by notifying the water supply office of K water to sampling progress. Chironomid sampling was conducted in bottom sediments and aquatic weeds at the water sources and DWTPs. Quantitative surveys of chironomids were conducted using a dredge (mesh size: 0.25 μm) and a Surber net (mesh size: 0.25 μm). A hand net was employed for qualitative surveys of chironomids. Sampling was performed in triplicate at each DWTP site. All samples were stored in 95% ethanol and transported to the laboratory for chironomid species classification. A YSI 63 handheld system (YSI Incorporated, Yellow Springs, Ohio, USA) was used to measure water quality factors, including water temperature, dissolved oxygen (DO), pH, and conductivity (EC), to assess the physicochemical characteristics of the water. Riverbed structures at the sampling sites of the DWTPs were also investigated.

## Morphological identification and DNA barcoding of chironomids from DWTPs

Morphological identification and DNA barcoding were performed on chironomids collected from DWTPs following established methods [1]. For morphological identification, the head and posterior appendage or the remaining trunk of each chironomid were dissected and mounted on slides. Taxonomic identification of each larva was then conducted by observing the mentum, mandible, labrum, antennae, and anal setae using a light microscope (Olympus, BX51, Shinjuku, Japan) [22, 23].

 For DNA barcoding, the remaining trunk of each chironomid was used for DNA extraction with the AccuPrep® Genomic DNA Extraction Kit (Bioneer, Daejeon, South Korea),

following the manufacturer's protocol. The purity and concentration of the extracted DNA were measured using a NanoDrop ND-2000 spectrophotometer (Implen, Munich, Germany). The polymerase chain reaction (PCR) to amplify the mitochondrial *COI* gene included an initial denaturation step at 94°C for 5 min, followed by 37 cycles at 95°C for 40 s, 57°C for 40 s, and 72°C for 50 s, with a final extension step at 72°C for 5 min, and was performed using *Accu-Power*® PCR PreMix & Master Mix (Bioneer, Daejeon, South Korea). The *COI* primers for PCR were: forward primer (5'-TTTCTACAAATCATAAGATA TTGG-3') and reverse primer (5'-TAAACTTCAGGGTGACCAAAAAATCA-3'). Purification of PCR products was carried out using the Solg™ Gel & PCR Purification Kit (Solgent, Daejeon, South Korea), and direct sequencing was conducted using an ABI 3730*xl* DNA Analyzer (Applied Biosystems, Massachusetts, USA).

### Phylogenetic tree analysis

Phylogenetic trees were constructed using the MEGA v10.2.4 program [24] with the Kimura-2 parameter (K2P) and maximum composite likelihood models. Bootstrap analysis was conducted with 1,000 replicates. The *COI* sequences of the chironomids found in DWTPs were compared with those in the NCBI database to construct the phylogenetic tree.

## Results

### Distribution of chironomid larvae in water intake and freshwater resources of DWTPs

A total of 180 individuals were collected from the water intake source (DY) of the Danyang DWTP in April 2024 (Fig 1A–1C). The chironomid larvae found at the DY site belonged to three subfamilies (Chironominae, Orthocladiinae, and Tanypodinae), with six species from six genera (Fig 2A). The collected Chironominae consisted of four species: *C. dorsalis*,

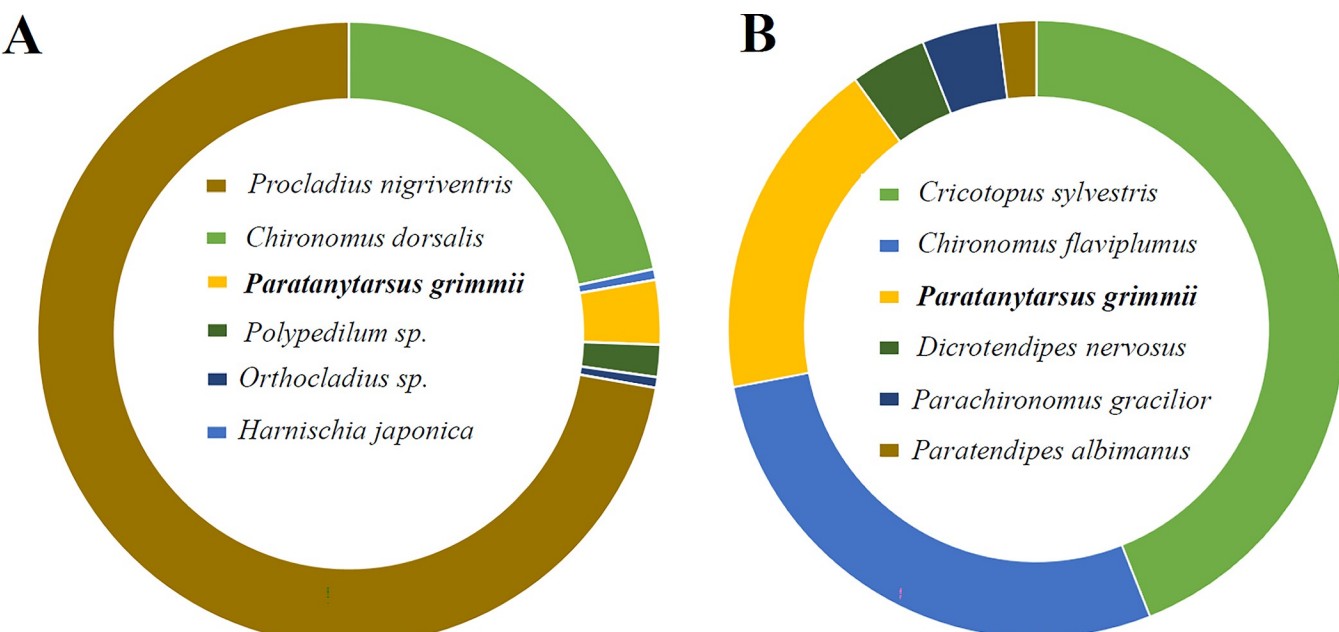

**Fig 2.** Distribution of chironomids at the DWTP sampling sites (A) DY site and (B) ND site. The chart illustrates the chironomid community composition at each DWTP, expressed as a percentage of the total collected individuals. Different colors represent distinct chironomid species identified at each DWTP.

*Polypedilum* sp., *Harnischia japonica*, and *P. grimmii*, the latter referred to as the parthenogenetic chironomid. Additionally, *Orthocladius* sp. from the Orthocladiinae subfamily and *Procladius nigriventris* from the Tanypodinae subfamily were also found at the DY sites. The dominant species among the chironomid larvae at the DY site was *P. nigriventris*, followed by *C. dorsalis* and *P. grimmii* larvae.

In the ND site, a total of 50 individuals were collected through water source sampling in April 2023 (Fig 1D–1F). The ND site serves as an indirect freshwater resource for the Busan Beomeo DWTP and Samgye DWTP [1]. The chironomid larvae found at the ND site belonged to two subfamilies (Chironominae and Orthocladiinae), with six species from six genera (Fig 2B). The collected Chironominae consisted of five species: *C. flaviplumus*, *Dicrotendipes nervosus*, *Parachironomus gracilior*, *Paratendipes albimanus*, and *P. grimmii*. The dominant species among the chironomid larvae at the ND site was *Cricotopus sylvestris* from the Orthocladiinae subfamily, followed by *C. flaviplumus*, *P. grimmii*, *D. nervosus*, *P. gracilior*, and *P. albimanus*.

### Water environments in freshwater sources of DWTPs

Intake source sampling site (DY) of the Danyang DWTP were as follows: physicochemical characteristics: water temperature, 17.8°C; DO, 11.3 mg L$^{-1}$; pH, 8.1; EC, 235.5 μs cm$^{-1}$; biological oxygen demand (BOD), 1.7 mg L$^{-1}$; suspended solids (SS), 8.5 mg L$^{-1}$; and turbidity, cloudy. The riverbed at the DY site was composed of 20% mud (<0.063 mm), 30% fine sand (0.063–2 mm), 15% pebbles (16–64 mm), and 35% small stones (64–256 mm). The water source sampling site (ND) exhibited the following physicochemical characteristics: water temperature, 18.6°C; DO, 10.45 mg L$^{-1}$; pH, 8.66; EC, 391 μs cm$^{-1}$; and turbidity, cloudy. The riverbed structure at the ND site consisted of 84% mud (<0.063 mm) and 16% fine sand (0.063–2 mm).

### Morphological characteristics of *P. grimmii* larvae found in sampling sites

The body length of *P. grimmii* larvae found at the water intake source site (DY) of Danyang DWTP (Fig 3) and the water source site (ND) of Nakdong River (Fig 4) was 5.0 ± 1.8 mm, with a light brown or beige coloration (Figs 3A and 4A). The head width was 180 ± 12 μm, with two symmetrically arranged eye points of equal size, arranged vertically (Figs 3B and 4B). The anal setae did not exhibit distinct features (Figs 3C and 4C). The *P. grimmii* larvae exhibited a mentum with 11 teeth: one median tooth (MT) and five pairs of lateral teeth (LT), arranged in a 5LT-1MT-5LT configuration. The median tooth was raised, whereas the lateral teeth were aligned and tapered downward (Figs 3D and 4D). The mandible contained one apical tooth, one dorsal tooth, and 2–3 inner teeth (Figs 3E and 4E). The ventromental plates were thin and wider than the mentum. The antennae were divided into five segments. The ring organ on the first segment was located in the lower center of the segment, and the end of this segment had a blade approximately the length of the second segment. A style and two Lauterborn organs were located at the end of the second segment, and the pedicel connected to the Lauterborn organs was short. The Lauterborn organ exhibited a pointed end, extending to the middle of the second segment (Figs 3F and 4F).

### Phylogenetic analysis of *P. grimmii* among global *P. grimmii* species and chironomid species found in water resources of DWTPs

Fig 5 illustrates the phylogenetic relationships between the mitochondrial COI sequences of *P. grimmii* larvae first found in Korean DWTP sites and the *P. grimmii* COI sequences reported worldwide in the NCBI database. The sequence identity (SI) of the COI gene was 100% among three *P. grimmii* individuals from the DY site. Analysis of samples from five countries,

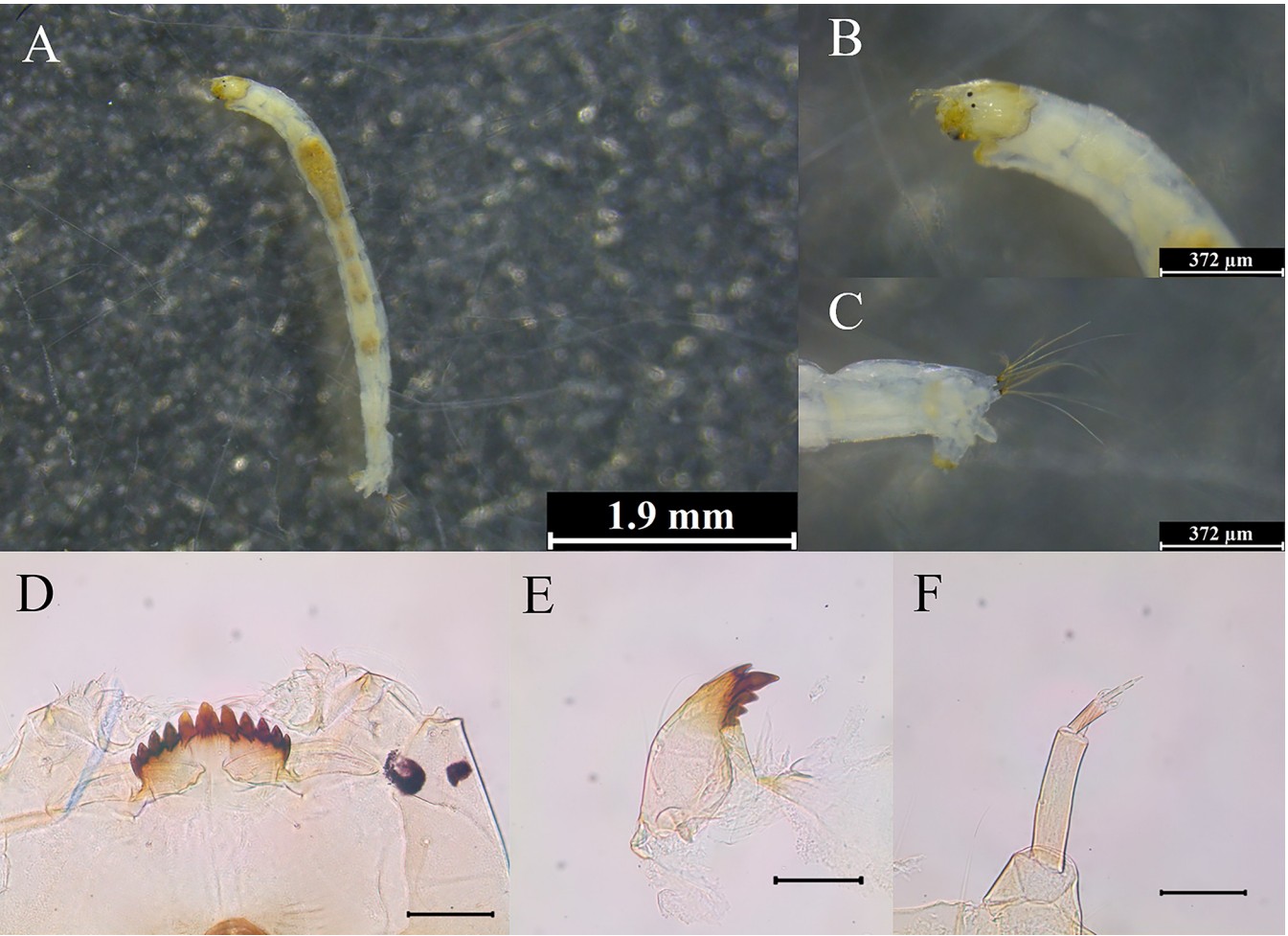

**Fig 3. Morphological identification of chironomid larvae collected from the DY site of the DWTPs.** (A) Overview of *P. grimmii* larvae, with detailed features including (B) head capsule, (C) anal setae, (D) mentum, (E) mandible, and (F) antennae. Scale bars are 1.9 mm for (A) and 372 μm for (B–F).

including Korea (DY site), Japan, Poland, Norway, and Germany, revealed a lack of mitochondrial COI diversity, with a pairwise distance value of 0.000 inferred from the nucleotide substitution of *P. grimmii* COI sequences. However, three *P. grimmii* individuals found at the ND site exhibited mitochondrial COI diversity, with pairwise distance values ranging from 0.005 to 0.008, inferred from the nucleotide substitution of *P. grimmii* COI sequences. Although the SI of other COI sequences from Australia and Canada in the NCBI database was over 99%, the sequence variation in *P. grimmii* from the ND site was relatively high compared to *P. grimmii* individuals reported worldwide in the NCBI database and the three *P. grimmii* individuals from the DY site.

The phylogenetic tree of chironomid larvae collected from the water intake and freshwater sources of DWTPs, along with related chironomid species from the NCBI database, is presented in Fig 6. A total of 694 individuals were collected from these sources in April. The chironomid larvae found at the DWTP sampling sites belonged to four subfamilies: Chironominae (7 species), Orthocladiinae (2 species), Diamesinae (1 species), and Tanypodinae (2 species), encompassing 12 species from 11 genera, as determined through morphological identification and *COI* mitochondrial DNA analysis. *Paratanytarsus grimmii* individuals

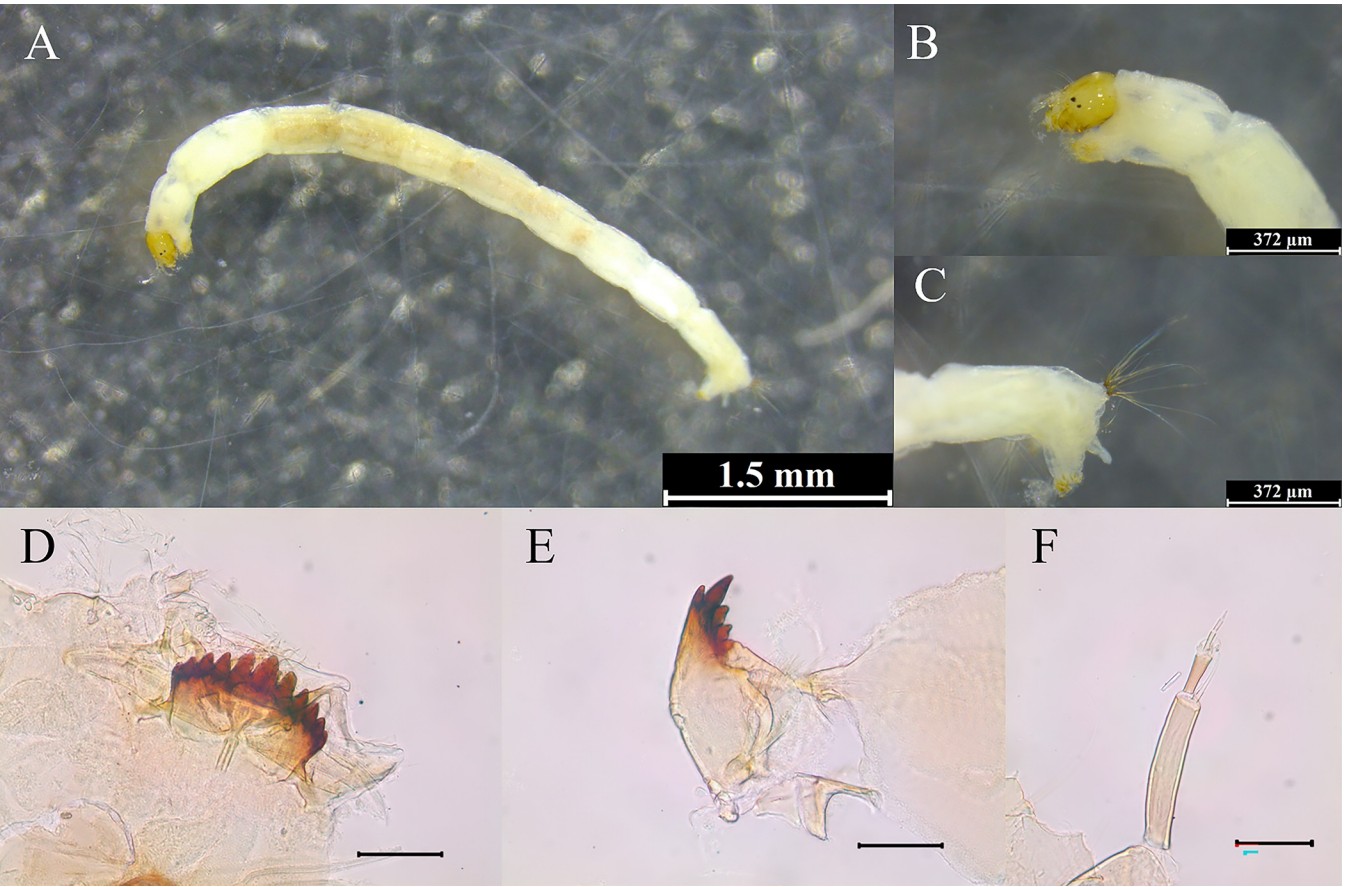

**Fig 4. Morphological identification of chironomid larvae collected from the ND site of the Nakdong River.** (A) Overview of *P. grimmii* larvae, with detailed features including (B) head capsule, (C) anal setae, (D) mentum, (E) mandible, and (F) antennae. Scale bars are 1.5 mm for (A) and 372 μm for (B–F).

belong to the Chironominae subfamily, and the collected Chironominae consisted of six genera and seven species: *C. dorsalis*, *C. flaviplumus*, *Microchironomus* sp., *Paratendipes* sp., *Polypedilum* sp., *Tanytarsus shoudigitatus*, and *P. grimmii*. In the phylogenetic tree, *P. grimmii* species were closely related to *T. shoudigitatus* species within the Chironominae subfamily.

## Discussion

Ensuring the quality of water exiting from DWTPs is of paramount importance due to its significant impact on public health for millions of people relying on drinking water [25]. Although the primary step in DWTPs involves disinfection to eliminate microbial contamination, issues related to the inflow of chemicals and microplastics into DWTPs have been reported, posing a threat to the safety of drinking water [25–27]. Furthermore, the detection of macroinvertebrate larvae in DWTPs further highlights the safety concerns associated with drinking and tap water [13, 14]. Freshwater environments such as lakes, rivers, and streams are key habitats for a diverse range of aquatic macroinvertebrates [10, 28, 29]. Therefore, the presence of chironomids in aquatic ecosystems is highly likely. However, in DWTP networks, massive occurrence of macroinvertebrates presents a significant challenge for managing water quality. The development of organisms can negatively impact drinking water quality due to their potential for parthenogenic reproduction, the risk of harmful microbial communities

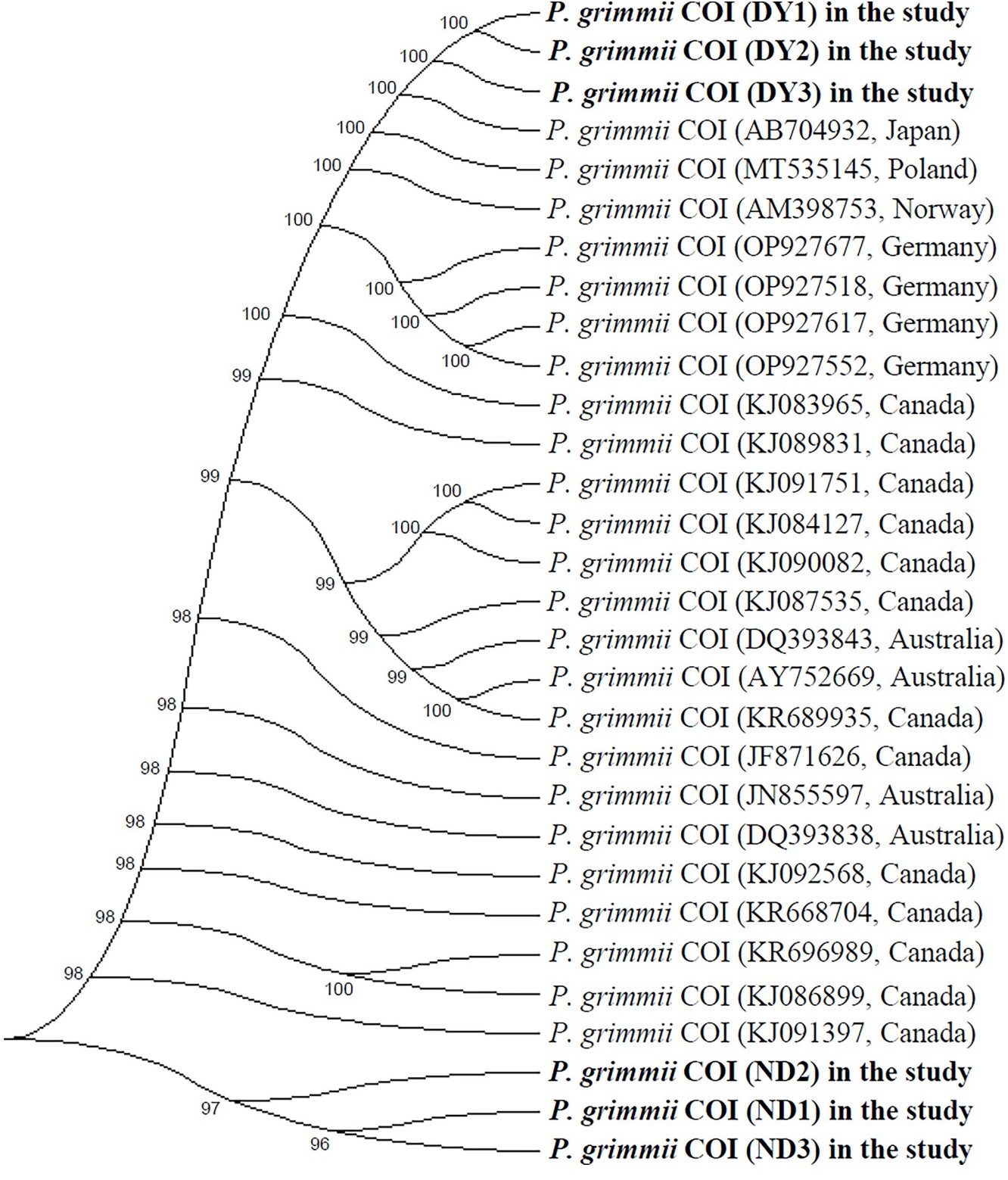

**Fig 5. Phylogenetic tree of *COI* genes from six *P. grimmii* specimens found in DWTPs in Korea, compared with other *P. grimmii COI* sequences reported in the NCBI GenBank database (representing twenty-four individuals from Japan, Poland, Norway, Germany, Canada, and Australia).** The phylogenetic tree was constructed using neighbor-joining analysis with MEGA 4.0 software. Bootstrap values are indicated at the nodes (1,000 replicates).

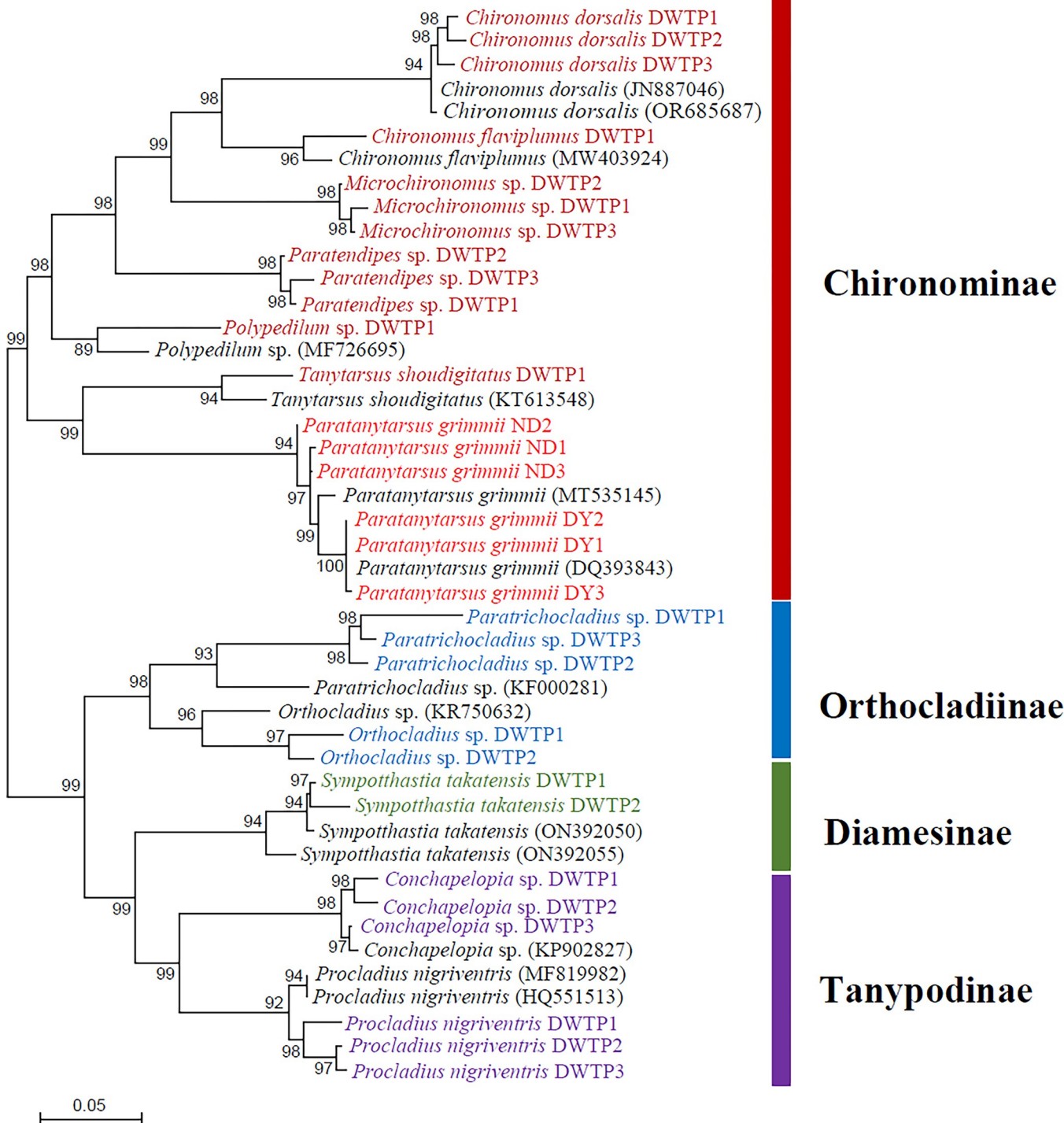

**Fig 6. Phylogenetic analysis of *COI* genes from chironomid larvae found in DWTPs, compared with other chironomid *COI* sequences reported in the NCBI GenBank database.** The phylogenetic tree was constructed using neighbor-joining analysis with MEGA 4.0 software. Bootstrap values are indicated at the nodes (1,000 replicates). The shading of the bars indicates chironomid species belonging to Chironominae (red), Orthocladiinae (blue), Diamesinae (green), or Tanypodinae (purple).

forming during the fouling of DWTP walls, and aesthetic issues arising from the presence of visible larvae in tap water and household water filters [17].

Larvae of the chironomid *P. grimmii* can reproduce parthenogenetically, with virgin reproduction occurring in the pupal stage of female or pharate females. The occurrence of *P. grimmii* was first reported in 1941, and it was detected in the drinking water distribution systems (DWDSs) of Northern Germany in 2019. It was also found in a water hydrant of the DWDSs in 2021 [17, 18]. However, there have been no reports of *P. grimmii* in water intake sources, freshwater sources of DWTPs, DWTP networks, or water hydrants in Korea [1, 13, 14], despite continuous monitoring of chironomid communities in DWTP water sources in Korea since 2020. This study is the first to report the presence of *P. grimmii* larvae in the water intake source and freshwater resource of DWTPs in Korea. However, data on the morphological characteristics and DNA barcode of the chironomid *P. grimmii* are scarce [16, 17, 30].

Pattern analysis of mitochondrial and microsatellite variation revealed extremely low mitochondrial diversity (<0.14%) in the widely distributed triploid chironomid species *P. grimmii* across Japan, England, Germany, Australia, and Canada. The microsatellite variation also indicated local endemism [16]. Interestingly, *P. grimmii* larvae sampled in this study exhibited different mitochondrial diversities (SI) depending on their regional habitats within Korea. Specifically, *P. grimmii* larvae from the ND site displayed relatively high diversity compared to all *P. grimmii* individuals reported in the NCBI database, whereas larvae from the DY site showed no diversity when compared to other *P. grimmii* individuals from DWTPs and the NCBI database. Furthermore, the morphological characteristics of *P. grimmii* larvae, including features of the mentum, mandible, ventromental plates, and antennae, were distinct from those of other chironomid larvae. Phylogenetic analysis indicated that parthenogenetic *P. grimmii* larvae coexist with other chironomid species from four subfamilies in the water intake sources of DWTPs. Fortunately, there have been no reports of *P. grimmii* larvae being found in the processing stages of DWTPs or in household water filters in Korea. However, the presence of this species in water intake and freshwater sources suggests a potential for future detection in the DWTP processing stages. Parthenogenetic triploid species such as *P. grimmii* often have the potential to spread rapidly across vast geographical areas [17].

## Conclusions

Non-biting midges (Diptera: Chironomidae), are the most ubiquitous aquatic insects in freshwater environments used as water sources for DWTPs. The presence of chironomid larvae in DWTPs is frequently reported during the summer season in Korea. This study investigated the distribution of chironomid larvae collected from the intake water source (DY) and freshwater resources (ND) of DWTPs in South Korea. Notably, *P. grimmii* larvae were identified for the first time at these sampling sites. The chironomid larvae collected from the DWTPs belonged to four subfamilies: Chironominae, Orthocladiinae, Diamesinae, and Tanypodinae, encompassing 12 species across 11 genera, as determined by morphological identification and *COI* mitochondrial DNA analysis. Morphological characteristics of *P. grimmii* larvae (body length: 5.0 ± 1.8 mm, head width: 180 ± 12 μm) were identified using features such as the mentum (a 5LT-1MT-5LT configuration), mandible, and antennae. In DNA barcoding and phylogenetic analysis, the mitochondrial *COI* sequences of *P. grimmii* from the DY site in Korea and from the NCBI database (representing Japan, Poland, Norway, and Germany) exhibited no mitochondrial *COI* diversity, with a pairwise distance value of 0.000. Conversely, *P. grimmii* individuals from the ND site in Korea displayed a mitochondrial *COI* diversity with pairwise distance values ranging from 0.005 to 0.008. The genetic and morphological data on the triploid parthenogenetic *P. grimmii* obtained through this study can provide valuable information

for biomonitoring efforts aimed at detecting the presence of this species in freshwater resources and DWTPs, thereby supporting the provision of high-quality drinking and tap water from DWTPs.

## Supporting information

**S1 File.**
(XLSX)

## Acknowledgments

Comments and suggestions of three anonymous reviewers and an editor improved our manuscript, for which we are grateful.

## Author Contributions

**Conceptualization:** Jae-Won Park, Kiyun Park, Ihn-Sil Kwak.

**Data curation:** Jae-Won Park, Kiyun Park, Ihn-Sil Kwak.

**Formal analysis:** Jae-Won Park.

**Funding acquisition:** Ihn-Sil Kwak.

**Investigation:** Jae-Won Park, Kiyun Park.

**Methodology:** Jae-Won Park.

**Project administration:** Ihn-Sil Kwak.

**Resources:** Jae-Won Park, Kiyun Park.

**Software:** Kiyun Park.

**Supervision:** Ihn-Sil Kwak.

**Validation:** Jae-Won Park, Kiyun Park, Ihn-Sil Kwak.

**Visualization:** Jae-Won Park, Kiyun Park.

**Writing – original draft:** Jae-Won Park, Kiyun Park.

**Writing – review & editing:** Kiyun Park, Ihn-Sil Kwak.

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
