## [Decision Letter · Decision Letter 0]

25 Oct 2024

PONE-D-24-33402

First report of a major management target species, parthenogenetic chironomid Paratanytarsus grimmii (Diptera: Chironomidae) larvae, in drinking water treatment plants (DWTPs) in South Korea

PLOS ONE

Dear Dr. Kwak,

Thank you for submitting your manuscript to PLOS ONE. After careful consideration, we feel that it has merit but does not fully meet PLOS ONE’s publication criteria as it currently stands. Therefore, we invite you to submit a revised version of the manuscript that addresses the points raised during the review process.

This is an interesting and well written submission. However, I have some concerns which were pointed out in a partial review of the work (e.g., reviewer 2 did not provide a full review as these issues where deemed the primary issues). I agree that these need to be addressed and suggest that main themes could framed and explained (see comment below) more clearly. If these can be addressed, I feel the work would be more suitable for publication. 

We look forward to receiving your revised manuscript.

Kind regards,

Michael A. Chadwick, PhD

Academic Editor

PLOS ONE

“This work was supported by a grant from the National Institute of Biological Resources (NIBR), funded by the Ministry of Environment (MOE) of the Republic of Korea (NIBR202220202) and the National Research Foundation of Korea, South Korea, funded by the Korean Government (NRF-2018-R1A6A1A-03024314), as well as the Korean Environment Industry & Technology Institute (KEITI) through the Aquatic Ecosystem Conservation Research Program funded by the Korean Ministry of Environment (MOE) (2021003050001).”

“This work was supported by a grant from the National Institute of Biological Resources (NIBR), funded by the Ministry of Environment (MOE) of the Republic of Korea (NIBR202220202) and the National Research Foundation of Korea, South Korea, funded by the Korean Government (NRF-2018-R1A6A1A-03024314), as well as the Korean Environment Industry & Technology Institute (KEITI) through the Aquatic Ecosystem Conservation Research Program funded by the Korean Ministry of Environment (MOE) (2021003050001).”

“This work was supported by a grant from the National Institute of Biological Resources (NIBR), funded by the Ministry of Environment (MOE) of the Republic of Korea (NIBR202220202) and the National Research Foundation of Korea, South Korea, funded by the Korean Government (NRF-2018-R1A6A1A-03024314), as well as the Korean Environment Industry & Technology Institute (KEITI) through the Aquatic Ecosystem Conservation Research Program funded by the Korean Ministry of Environment (MOE) (2021003050001).”

Additional Editor Comments:

This is an interesting submission, but needs to address to major concerns before being considered for publication. 1) Ptny. grimmii is not a obligate parthenogenetic species, but a cyclical one, perhaps by thermo-dependent regulator, the mechanism is not clear now. At certain condition, some male can be produced. 2) Either adults or larva, chironomid is not a pest insect, but a nuisance insect, casting limited effluence on public health. If these issues can be addressed, I feel the work would be improved and then likely suitable for publication.

Reviewers' comments:

Reviewer's Responses to Questions

**Comments to the Author**

1. Is the manuscript technically sound, and do the data support the conclusions?

Reviewer #1: Yes

2. Has the statistical analysis been performed appropriately and rigorously? 

Reviewer #1: Yes

3. Have the authors made all data underlying the findings in their manuscript fully available?

Reviewer #1: Yes

4. Is the manuscript presented in an intelligible fashion and written in standard English?

Reviewer #1: Yes

5. Review Comments to the Author

Reviewer #1: The paper is well structured, with new interesting information about intraspecific variability of a Chironomid species of high practical interest, living in tap waters.

Only minor adjustments are needed: e. g. Fig. (abbreviated) and Figure (not abbreviated) citations are mixed;

at row 257 a t before P. grimmi must be deleted

6. PLOS authors have the option to publish the peer review history of their article (what does this mean?). If published, this will include your full peer review and any attached files.

Reviewer #1: **Yes: **Bruno Rossaro

---

## [Author Response · Author response to Decision Letter 0]

20 Nov 2024

Reviewers' comments:

Reviewer's Responses to Questions

Comments to the Author

1. Is the manuscript technically sound, and do the data support the conclusions?

Reviewer #1: Yes

 � Thank you for the comment of reviewer #1.

2. Has the statistical analysis been performed appropriately and rigorously?

Reviewer #1: Yes

 � Thank you for the comment of reviewer #1.

3. Have the authors made all data underlying the findings in their manuscript fully available?

Reviewer #1: Yes

 � Thank you for the comment of reviewer #1.

4. Is the manuscript presented in an intelligible fashion and written in standard English?

Reviewer #1: Yes

 � Thank you for the comment of reviewer #1.

5. Review Comments to the Author

Reviewer #1: The paper is well structured, with new interesting information about intraspecific variability of a Chironomid species of high practical interest, living in tap waters.

Thank you for the comment of reviewer #1.

Only minor adjustments are needed: e. g. Fig. (abbreviated) and Figure (not abbreviated) citations are mixed; at row 257 a t before P. grimmi must be deleted

Lines 201 and 217, we have revised as the format of “Fig. (abbreviated)” citations in the revised manuscript.

Line 264, we have deleted and revised the typo error in the revised manuscript.

6. PLOS authors have the option to publish the peer review history of their article (what does this mean?). If published, this will include your full peer review and any attached files.

Do you want your identity to be public for this peer review? For information about this choice, including consent withdrawal, please see our Privacy Policy.

Reviewer #1: Yes: Bruno Rossaro

 � Thank you for the comment of reviewer Dr. Bruno Rossaro.

Additional Editor Comments:

This is an interesting submission, but needs to address to major concerns before being considered for publication. 1) Ptny. grimmii is not a obligate parthenogenetic species, but a cyclical one, perhaps by thermo-dependent regulator, the mechanism is not clear now. At certain condition, some male can be produced. 2) Either adults or larva, chironomid is not a pest insect, but a nuisance insect, casting limited effluence on public health. If these issues can be addressed, I feel the work would be improved and then likely suitable for publication.

We agree with the editor’s comments. The chironomid P. grimmii is not an obligate parthenogenetic species and a pest insect. 

We have revised the issue in the revised manuscript.

Lines 2-4, we have revised the title as the below.

Before: First report of a major management target species, parthenogenetic chironomid Paratanytarsus grimmii (Diptera: Chironomidae) larvae, in drinking water treatment plants (DWTPs) in South Korea

After revision: First report of a major management target species, chironomid Paratanytarsus grimmii (Diptera: Chironomidae) larvae, in drinking water treatment plants (DWTPs) in South Korea

Lines 25, 39, 79, 81, 93, 155, 240, 244, We have deleted the word “pest” in the revised manuscript.

Line 27, we have revised the word from “risk” to “possibility” in the revised manuscript.

Lines 39-40, we have revised the word from “pest” to “key management target” in the revised manuscript. 

Line 79, we have deleted the “and an apomictic pest” in the revised manuscript.

Line 240, we have revised the word from “seriously” to “negatively” in the revised manuscript.

Thanks in advance. 

Yours sincerely,

Ihn-Sil Kwak, Ph.D.

Dept. of Ocean Integrated Science, 

Chonnam National University, Chonnam, 550-749, Korea.

E-mail address: iskwak@chonnam.ac.kr/inkwak@hotmail.com

---

## [Decision Letter · Decision Letter 1]

26 Nov 2024

First report of a major management target species, chironomid Paratanytarsus grimmii (Diptera: Chironomidae) larvae, in drinking water treatment plants (DWTPs) in South Korea

PONE-D-24-33402R1

Dear Dr. <!--StartFragmentIhn-Sil Kwak<!--EndFragment,

We’re pleased to inform you that your manuscript has been judged scientifically suitable for publication and will be formally accepted for publication once it meets all outstanding technical requirements.

Kind regards,

Sanja Puljas

Academic Editor

PLOS ONE

Additional Editor Comments (optional):

Reviewers' comments:

Reviewer's Responses to Questions

**Comments to the Author**

1. If the authors have adequately addressed your comments raised in a previous round of review and you feel that this manuscript is now acceptable for publication, you may indicate that here to bypass the “Comments to the Author” section, enter your conflict of interest statement in the “Confidential to Editor” section, and submit your "Accept" recommendation.

Reviewer #1: All comments have been addressed

2. Is the manuscript technically sound, and do the data support the conclusions?

Reviewer #1: Yes

3. Has the statistical analysis been performed appropriately and rigorously? 

Reviewer #1: Yes

4. Have the authors made all data underlying the findings in their manuscript fully available?

Reviewer #1: Yes

5. Is the manuscript presented in an intelligible fashion and written in standard English?

Reviewer #1: Yes

6. Review Comments to the Author

Reviewer #1: The paper is surely interesting and merits publication.

But one item must be better discussed before the publication of the paper.

The question is about the "parthenogenetic" status of the species. The species was considered "true" parthenogenetic as

supported by a rich literature;

Schneider, A. 1885 Chironomus Grimmii und seine Parthenogenesis. Zool. Beitr. 1(3): 301.

Langton, P. H., Cranston, P. S., Armitage, P. 1988 The parthenogenetic midge of water supply systems, Paratanytarsus grimmii (Schneider) (Diptera: Chironomidae) Bull. ent. Res. 78: 317-328.

Porter, D. L. 1971 Oogenesis and chromosomal heterozygosity in the thelytokous midge, Lundstroemia parthenogenetica (Diptera, Chironomidae) Chromosoma 32: 332-342.

Kondo, S. 1998 Life history characteristics of Paratanytarsus grimmii Schneider (Chironomidae) from the Yamazaki River, Japan In: Ismay, J. W. (ed.): Abstr. 4th Int. Congr. Dipterol., Oxford: 109.

Gagliardi, B. S., Long, S. M., Pettigrove, V. J., Hoffmann, A. A. 2015 The parthenogenetic cosmopolitan chironomid, Paratanytarsus grimmii, as a new standard test species for ecotoxicology: Culturing methodology and sensitivity to aqueous pollutants. Bull. Envir. Contam. Toxicol. 95(3): 350-356

Olsen, A., Bale, J. S., Leadbeater, B. S. C., Callow, M. E., Holden, J. B. 2003 Developmental thresholds and day-degree requirements of Paratanytarsus grimmii and Corynoneura scutellata (Diptera: Chironomidae): two midges associated with potable water treatment. Physiol. Ent. 28(4): 315-322.

Olsen, A., Leadbeater, B. S. C., Callow, M. E., Holden, J. B. and Bale, J. S. 2009 The origin and population dynamics of annually re-occurring Paratanytarsus grimmii (Diptera: Chironomidae) colonising granular activated carbon (GAC) adsorbers used in potable water treatment Bull. ent. Res. 99: 643-651.

Very recently this paper was submitted to Zootaxa and is actually "in press":

First description of male from parthenogenesis in a chironomid species Paratanytarsus grimmii (Schneider, 1885) (Diptera: Chironomidae)

ATSUYA KODAMA1, HIDETOSHI SAITO2,3 & KOICHIRO KAWAI2,4

1: Tsuruga City, Fukui Prefecture, Japan. e-mail: dip.tanytarsus.ko.25@gmail.com

2: Laboratory of Aquatic Ecology, Graduate School of Integrated Science for Life, Hiroshima University, 1-4-4 Kagamiyama, Higashi-Hiroshima City, Hiroshima, Japan

3: e-mail: saito@hiroshima-u.ac.jp

4: e-mail: kawagogi@hiroshima-u.ac.jp

descibing the adult males of the species.

I suggest to contact these authors to agree an addition to your paper which definitely clarify the "parthenogentic status" of the species

7. PLOS authors have the option to publish the peer review history of their article (what does this mean?). If published, this will include your full peer review and any attached files.

Reviewer #1: **Yes: **Bruno Rossaro

---

## [Editor Report · Acceptance letter]

31 Dec 2024

PONE-D-24-33402R1 

PLOS ONE

Dear Dr. Kwak, 

I'm pleased to inform you that your manuscript has been deemed suitable for publication in PLOS ONE. Congratulations! Your manuscript is now being handed over to our production team.

Kind regards, 

on behalf of

Dr. Sanja Puljas 

Academic Editor

PLOS ONE